# Gut Microbiota at the Intersection of Alcohol, Brain, and the Liver

**DOI:** 10.3390/jcm10030541

**Published:** 2021-02-02

**Authors:** Haripriya Gupta, Ki Tae Suk, Dong Joon Kim

**Affiliations:** Institute for Liver and Digestive Diseases, Hallym University, Chuncheon 24253, Korea; phr.haripriya13@gmail.com (H.G.); ktsuk@hallym.ac.kr (K.T.S.)

**Keywords:** alcohol, alcoholic liver disease, gut–liver–brain axis, bacterial metabolites

## Abstract

Over the last decade, increased research into the cognizance of the gut–liver–brain axis in medicine has yielded powerful evidence suggesting a strong association between alcoholic liver diseases (ALD) and the brain, including hepatic encephalopathy or other similar brain disorders. In the gut–brain axis, chronic, alcohol-drinking-induced, low-grade systemic inflammation is suggested to be the main pathophysiology of cognitive dysfunctions in patients with ALD. However, the role of gut microbiota and its metabolites have remained unclear. Eubiosis of the gut microbiome is crucial as dysbiosis between autochthonous bacteria and pathobionts leads to intestinal insult, liver injury, and neuroinflammation. Restoring dysbiosis using modulating factors such as alcohol abstinence, promoting commensal bacterial abundance, maintaining short-chain fatty acids in the gut, or vagus nerve stimulation could be beneficial in alleviating disease progression. In this review, we summarize the pathogenic mechanisms linked with the gut–liver–brain axis in the development and progression of brain disorders associated with ALD in both experimental models and humans. Further, we discuss the therapeutic potential and future research directions as they relate to the gut–liver–brain axis.

## 1. Introduction

Over the last few decades, maintaining a proper diet has proven difficult for many around the globe as social factors and a more sedentary lifestyle often lead to poor health choices [1]. Additionally, the consumption of alcohol is quite common in day to day life, causing liver disease worldwide, which accounts for approximately 4% of all deaths globally [2].

Alcoholic liver disease (ALD) consists of a broad spectrum from simple reversible steatosis to steatohepatitis, followed by fibrosis prominent to cirrhosis, eventually leading to hepatocellular carcinoma. Nevertheless, alcoholic cirrhosis (AC) is the most common cause of death in ALD [3,4]. From a pathophysiological perspective, multifactorial pathways are responsible for alcohol-induced steatosis, which is the early sign of liver injury in heavy-drinking adults. Oxidation of ethanol and its first metabolic product, acetaldehyde, produces several other cofactors that are involved in cellular redox reactions and are responsible for the enhancement of lipogenic gene expression, simultaneously lowering the expression of lipolytic genes. This action promotes lipid accumulation and decrease fatty acid oxidation in hepatocytes [5]. Moreover, liver insults can also be boosted by comorbidities, such as chronic viral hepatitis, non-alcoholic fatty liver, and epigenetic factors [6,7].

The primary methods by which the intestine communicates with the liver is through the hepatic portal vein, hepatic biliary system, and by using other potential intermediary substances for maintaining normal physiological conditions [8]. As a natural inhabitant of the intestine, gut microbiota serves as invaluable contributors to gut health as well as in the pathophysiology of numerous diseases [9,10,11]. Alcohol ingestion alters various metabolic and biochemical processes, which are further aggravated by alcohol overconsumption. This ultimately leads to a series of hepatocellular changes, culminating in cirrhosis and hepatocellular carcinoma [3,12].

It appears that a variety of factors responsible for the modulation of ALD including as heredity, sex, and alcohol misuse are not sufficient to lead to liver disease by themselves. The disturbance of the intestinal microbiota is also important and has been identified recently as a potentially modifiable therapeutic target in the progression of ALD [4,13,14]. Moreover, the gut microbiome is also associated with obesity and inflammatory gastrointestinal disorders [15,16]. As for the bacteria’s own growth and cellular processes, they utilize ingested dietary constituents producing numerous metabolites that impact the host’s health and also pose potential disease risks [17,18,19].

The gut microflora and its metabolites have been shown to play a role in altering the function of the enteric nervous system (ENS). There is an increasing number of studies that connect the gut microbiome to the function and growth of the central nervous system (CNS), which is presently a new, anticipated shift in field of neurology. Some studies suggests that the composition of gut microbiota and brain disorders such as depression and chronic stress are intertwined and termed as the “gut–brain axis.” This field needs further investigation, especially considering its potential therapeutic options [20,21].

In recent years, scientists have validated the gut microbiome’s role in maintaining normal physiology and verified the host–biome symbiosis. Disrupting the microbiome creates a burden on the host’s normal physiological functions, thus contributing to disease progression. For such instances, scientists are developing new models to study gastrointestinal-associated liver diseases and the pathways facilitating inter-organ interactions during biological events. Animal-based preclinical studies [22,23,24], human clinical trials, and artificial-intelligence-guided omics studies [25,26,27,28] are ongoing, which have begun to elucidate the biological and cellular events that occur at the gut–liver inter-junctions [4]. Additionally, research into the gut microbiome has evoked a new paradigm of the gut–brain axis communication, demonstrating that gut microbes and the brain are closely linked in the bidirectional functions of neurons in CNS [29]. Herein, we reviewed the biological intersection of liver disease and the gut microbiota in concurrence with the third intersection, “the brain.” Additionally, we explored the hypothesis that gut bacteria are essential contributors to the progression of ALD, affecting mental health and causing disharmony between the host’s internal systems.

## 2. Communication between the Gut, Liver, and Brain in Alcoholic Liver Disease

The gut microbiota functions as an eminent yet natural source of metabolites, bioactive molecules, and endotoxins that regulate not only gastrointestinal physiology but also other organs including the brain, kidney, and cardiovascular system as well [30,31,32]. These microbes not only support inter-organ communication, but they are also a determining factor in triggering pathophysiological changes in many disease [33]. The anatomy of the liver provides a bidirectional link to the intestines through the hepatic portal system, which carries gut-derived metabolic products directly to the liver. Additionally, the biliary tracts and systemic circulation provide a platform for the liver to communicate back with the gut, the liver release bile acids and other bioactive molecules, which act as a feedback mechanism on the gut from the liver [34].

Chronic alcohol consumption changes the microbial composition, which insults the gut mucosal barrier and thus compromises gut homeostasis, which was maintained by the segregation of microbiota and the host’s immune cells [35]. Meanwhile, the CNS also communicates with the gut through the gut–brain axis to facilitate a physiological sense of the host’s body. This bidirectional communication is mediated by immune cells and hormones passed through blood–brain barrier (BBB) via parasympathetic neural activity [36].

### 2.1. Alcohol and Gut–Liver Interaction

Alcohol use disorder (AUD) and chronic alcohol drinking are pervasive globally and cause untold economic and physiological damage [37]. Alcohol abuse has been linked with changes in microbial composition affecting intestinal permeability, which exposes the liver to deleterious events in patients with ALD [38]. Moreover, this direct interaction between the gut and liver provides a major pathway for the development and progression of ALD through the gut–liver axis. Such progression is supported by various factors and including bile acids and their conjugates resulting from bacterial action in the intestine and enterohepatic circulation back to the liver via the hepatic portal vein [35,39]. The liver is highly sensitive to the end products of bacterial metabolism, and this strong responsiveness is likely to affect liver functions when evoked, impacting the modulation of microbiota and physiology of the host [40].

Lipopolysaccharide (LPS), an endotoxin produced by Gram-negative bacteria, is one of the main factors in the pathogenesis of ALD. Mean levels of serum LPS are positively related with the stages of ALD in previous studies [41,42]. The toll-like receptor 4 (TLR4) pathway is activated when LPS complex with LPS-binding protein binds to TLR4 [43]. This downstream signaling pathway activates Kupffer cells and the release of cytokines causing hepatocyte damage [44,45,46,47]. In addition to LPS, other bacterial products including peptidoglycan, lipoteichoic acid, porin, and flagellin can also translocate from the gut to portal tracts and liver. Bacterial particles were also observed in the circulating blood [48,49].

#### 2.1.1. Bacterial Metabolites

Short-chain fatty acids (SCFAs) and secondary bile acids (BAs) are the two major types of metabolites produced by gut microbiota [50]. Secondary BAs are biochemically modified bacterial metabolites (primary BAs), whereas SCFAs are produced from dietary components by the gut microbiota.

Being amphipathic in nature, BAs are pleiotropic signaling molecules that are synthesized in the liver and undergo massive bioconversion when released into the intestine. This serves an important role in nutrient absorption as the enzymatic actions of gut microbiota facilitate the absorption of lipid and lipophilic vitamins from dietary components [51]. More than 95% to total BAs are reabsorbed by active transport and trans-diffused through enterocytes, secreting into the liver again through the liver sinusoid via hepatic portal circulation. This entero-hepatic cycle is strictly monitored by factors in the gut lumen, intestinal mucosa, hepatocytes, and by local and systemic inflammation. This regulation maintains the intestinal BA at a level required to meet intestinal demands, which determines the efficacy of the entero-hepatic cycle and keeps vigilance on small intestinal bacterial overgrowth. Additionally, BA regulation also plays a vital role in maintaining whole-body lipid and sterol homeostasis [52,53,54,55]. While BAs pooled in the gut to meet demand, this pool later serves as major substrate for the bacterial biotransformation of primary BA to secondary BA in the colon. Chronic alcohol drinking has been associated with an increased secondary BA to primary BA ratio in AC patients [56].

In previous studies, SCFAs (acetate, butyrate, and propionate) have been shown not only to maintain the intestinal epithelial barriers by providing energy to enterocytes but also to promote anti-inflammatory activity in the intestine through immune-modulation [57]. Chronic alcohol drinking has been shown to impair the production of SCFAs through abnormal microbiota displaying reduced biosynthetic activity, and consequently, a reduction in the SCFA fecal content was observed in AC patients [58]. The abnormal bacterial metabolites observed in AUD are a consequence of shifts in the bacterial community composition across taxonomic levels. In the stool of patients with AC, beneficial taxa within the phyla Firmicutes and Bacteroidetes such as Lachnospiraceae, *Roseburia*, Ruminococcaceae, *Blautia*, and Bacteroidaceae are reduced in abundance; opposingly, organisms in the phyla Protobacteria, Bacteroidetes, and Firmicutes, including Enterobacteriaceae, Porphyromonadaceae, and Streptococcaceae are drastically increased in abundance in AC stool samples [59,60].

#### 2.1.2. Microbe-Associated Molecular Patterns

Disruption of the intestinal epithelial barrier allows the microbe-associated molecular patterns (MAMP)s to translocate to the extraintestinal space and circulate through systemic circulation into the liver, triggering pathophysiological conditions [61,62]. MAMPs such as lipopolysaccharides (LPS), peptidoglycan, or bacterial DNA, serve as ligands for pattern-recognition receptors on Kupffer cells [63] and hepatic stellate cells, [64] inducing a proinflammatory response. Toll-like receptors, pattern-recognition receptors, are the primary receptors mediating the inflammatory responses required for pathogen extermination. However, continuous exposure to such stimuli leads to rigorous activation of downstream proinflammatory signaling mechanisms, inducing inflammation as well as promoting fibrosis via activation of hepatic stellate cells. This mechanism was confirmed by several studies that provided evidence demonstrating increased blood endotoxin level (LPS and peptidoglycans) in ALD patients. These data further support the hypothesis that the translocation of endotoxin from the intestine to the liver, via either hepatic portal or systemic circulation, is associated with disease severity throughout the different stages of ALD [65,66]. Additionally, a high blood cytokines level and increased circulating bacterial DNA have been well correlated to the seriousness and progression of alcoholic hepatitis (AH) [38,67].

Therefore, it seems reasonable to assess whether these prompt proinflammatory responses are initiated by bacterial products translocation through the intestinal barrier. In addition, gut microbiota can metabolize the physiologically vital amino acid tryptophan into indole and its derivatives; this process can limit the availability of tryptophan [68], and recently this mechanism was thoroughly reviewed by Beatriz et al., where they affirmed the dysregulation of tryptophan metabolism in alcohol-related liver diseases [69].

Furthermore, disruption of circadian rhythm or mutation in circadian gene distinctly affects intestinal permeability [70], and this alteration in gene expression markedly worsens alcohol-induced gut dysbiosis, hepatic injury, and hepatic inflammation as showed in vitro and in vivo studies [71,72].

### 2.2. Alcohol and Gut–Brain Interaction

The ecological system of the gut microbiome is very vast and not limited to the gastrointestinal system. The composition, relative abundance, and bioactivity of gut microbes not only influence gut and liver function but also have consequences in brain function, behavior, and mood as well [73,74]. Gut microbiota have emerged in recent investigations as a key regulators of neuro-development and behavior in brain disorders.

A number of previous studies were performed aiming to elucidate the underlying mechanisms of gut microbiota as they relate to brain disorders via ENS and metabolic pathways. Previous research has suggested that bacterial metabolites can enter the brain through the BBB via the sensory nerves that innervate the gut. Most of the direct and indirect pathways including the bidirectional vagal-to-brain transmission, peripheral immune responses, tryptophan metabolism, hormone signaling, and bacterial metabolites such as SCFAs and other fermented by-products, have some level of influence on brain function [75,76,77,78].

Additionally, the gut–brain axis also serves as an important connection in liver diseases including AUDs and hepatic encephalopathy (HE) [17,79,80]. Leclercq et al. provided supporting clinical data that revealed increased gut permeability was correlated with increased anxiety, depression, and alcohol craving in patients with AUDs [65]. Another aspect influencing gut–brain interaction is central regulatory circadian mechanisms in the brain, which can alter the circadian clock in the gastrointestinal track, leading to susceptibility to intestinal pathophysiology. Since alcohol use has an unsettling relationship with the circadian clock, this further exacerbates alcohol’s effect on intestinal barrier integrity and has a potential role in liver and brain injuries [70,81,82]. Considering the nature of alcoholism, strategies to prevent recurrence after withdrawal are often ineffective, and current management approaches require serious reconsideration; focusing on the microbiome may yield potential therapeutic targets for reduction in the psychological symptoms associated with AUD-mediated gut–brain axis disharmony.

#### 2.2.1. Bacterial Metabolites

SCFAs are synthesized by gut microbiota by the fermentation of dietary components, which both generates energy for the host and provides a suitable growth environment for bacteria. SCFAs also stimulate blood flow in the colon and aid the integrity of the BBB when accessed through systemic circulation [83]. In animal study, intravenous or intraperitoneal administration of sodium butyrate after traumatic brain injury, can prevent BBB breakdown and promote neurogenesis [84,85,86]. SCFAs also tend to interact through sympathetic intervention of the superior cervical ganglia, shaping the physiological reflexes between the CNS and the gut [87]. It is therefore plausible that modulating SCFA levels could be useful in preventing brain dysfunction.

Like SCFAs, secondary BAs such as deoxycholic acid have been detected in the CNS, modulating BBB permeability via disruption of the tight junctions [88]. Another bacterial metabolite, trimethylamine, and its catalytic oxidation product, trimethylamines-*N*-oxides, were found to be present in the CNS further supporting the fact that bacterial metabolites can cross the BBB [89].

#### 2.2.2. Neurotransmitters

The ENS in the gastrointestinal tract detects pathogens and acts against them by generating the necessary protective response. These intrinsic neuroglial circuits serve as gut–brain communications and sense the presence of bacterial metabolites and convey the information to the brain to initiate the appropriate response. In part, this response can include the activation of neurons that recruit immune cells to modulate local tissue inflammation, strongly linking the immune response with the gut–brain axis [90,91]. Additionally, germ-free mice showed decreased hyperexcitability in gut sensory neurons, further highlighting the communication between the gut and the brain [92]. Peripherally, endotoxins can stimulate the immune systems into releasing proinflammatory cytokines. If not resolved, these proinflammatory cytokines make their way through the bloodstream to disrupt the BBB, thus altering neurological functions and propagating a damaging cycle of brain inflammation [93].

Glutamate, an excitatory neurotransmitter in brain, is metabolized by *Lactobacillus* and *Bifidobacterium* to produce γ-aminobutyric acid (GABA), an inhibitory neurotransmitter in gut. GABA acts locally relaying information from the gut and alters vagal signaling to the brain. Intriguing findings from a recent clinical study, published by Kirsten et al. revealed brain GABA levels are inversely correlated the severity of ALD [94]. Bioconversion of tryptophan creates diverse microbial metabolites including tryptamine, indole, and indole derivatives. These metabolites may act as signaling molecules or ligands for the aryl hydrocarbon receptor, reducing the CNS inflammation and limiting disease severity [95]. The induced receptor activation in CNS cells could alter brain communication through the gut–brain axis in neuropsychiatric and neurodegenerative disorders [96], and it is likely that alcohol abuse exacerbates these effects [97,98].

### 2.3. Alcohol and Brain–Liver Interaction

The implications of alcohol abuse are well-documented in ALD patients and are reviewed thoroughly from time to time [99,100,101]. However, the connection between ALD-associated brain dysfunctions and communication via the gut–brain axis has gathered the attention of many researchers in recent years. Moreover, the communication within the gut–brain axis and the severity of HE has been of particular interest. HE is a brain disorder exemplified by altered brain function caused by liver insufficiency in acute liver failure or cirrhosis.

Ammonia produced by the gut microbiota is a crucial driver of HE and could be considered a typical paradigm of gut–brain–liver axis diseases [102]. When the liver fails to clear ammonia produced by gut microbiota, high levels of circulating ammonia reach the CNS where it is primarily handled by astrocytes. It is possible that the overload of ammonia in the CNS could lead to astrocyte senescence and initiate a cascade of neurological events causing the brain dysfunction seen in cirrhosis [103,104].

Chronic alcohol consumption has the outcome of dysbiosis leading to increased translocation of bacteria and harmful metabolites into blood stream that drives HE development and thus affects the alcohol–liver–brain relationship [79,97]. A study in ALD patients by Bajaj et al. demonstrated an association between cognitive dysfunction and HE-related cirrhosis. An increase in the relative abundance of Enterobacteriaceae, Lactobacillaceae, Alcaligenaceae, Porphyromonadaceae, and Streptococcaceae in combination with reduced relative abundances of Ruminococcaceae and Lachnospiraceae were demonstrated with HE patients. Additionally, Porphyromonadaceae, and Alcaligenaceae were positively correlated with poor cognition and inflammation in HE patients [105].

In another study, brain imaging was carried out to check neuronal function and integrity as it was related to microbial dysbiosis. While diffusion tensor imaging revealed altered neuronal integrity and edema that correlated positively with Porphyromonadaceae, magnetic resonance spectroscopy demonstrated hyperammonemia-related astrocyte dysfunction. The observed astrocytes dysfunction was positively correlated with Enterobacteriaceae, Streptococcaceae, Peptostreptococcaceae, and Lactobacillaceae. On the contrary, Ruminococcaceae, Lachnospiraceae, and Clostridiales XIV are negatively related with astrocyte dysfunction in HE patients [30]. Thus, the interaction of the brain with systemic endotoxemia, inflammatory mediators, and ammonia can worsen neuro-inflammation in AUDs [106]. However, HE-associated ALD clinical studies are lacking vital data, and more research is required in order to design effective treatments for HE in ALD patients.

## 3. The Gut–Liver Axis in Conjunction with the Brain as the Third Axis

Chronic use of alcohol is linked with several alterations in the gut, liver, and brain. The crosstalk between the gut and liver is increasingly recognized, enhanced by the parallel rise in the incidence of gastrointestinal diseases, liver diseases, and brain disorders [97,107,108]. Needless to say, alcohol-associated gut dysbiosis leads to increased circulation of pathobionts, which is mechanistically related to impaired cognitive functions and other changes within brain (Figure 1).

Dysregulation of metabolism, lower amino acid, and bioenergetic metabolites, and higher endotoxemia and toxic metabolites could reduce the induction of interleukin-22 and its positive regulation of the regenerating islet-derived 3 protein in intestinal Paneth cells that maintain the inner mucus membrane. Chronic alcohol consumption decreases regenerating islet-derived 3 expression, promoting bacterial and metabolite translocation, implicating this protein in local and systemic inflammation [109]. Additionally, liver overload with toxic metabolites leads to an increase in systemic load that reaches the brain via BBB alteration causing toxicity to brain cells [110]. These endotoxins also contribute to hepatocyte death and result in a fibrotic response in the liver [108]. These findings come with profound perturbations to the correlation between neuropsychiatric disorders and systemic inflammation, including depression and dementia [111,112].

In 2018, researchers at the scientific meeting reported the presence of gut bacteria in human brain tissue. The study has not yet been published, but it suggests that microbes might somehow be making their way into the brain, albeit skeptics abound [113]. Additionally, the direct linkage between the intestinal epithelium and enteric neurons through the vagus nerve serves as a local neural network responsible for transmitting signals to the brain in response to bacterial metabolite stimulation [114]. Systemic divergence may prime the vagus nerve to renounce or modify its neuroprotective afferent signals, or to stimulate vagal signals that affect brain function. A dysregulation in vagal signaling could result in increased neuroinflammation as was observed upon alcohol withdrawal and persisted during chronic alcohol feeding [115].

It is very likely that any pro-inflammatory responses involve microglial cell activation, resident macrophages, and the subsequent recruitment of macrophages to the brain. This inflammatory response is also expected to worsen the pathological condition in liver disease following the decline of brain functions [116]. Therefore, under vagal signals from the gut due to microbial imbalance, the humoral pathway of bioactive molecules to the brains are expected to be enhanced. Moreover, alcohol-induced dysbiosis increases the neuroinflammation directly or through nutritional deficiencies that deteriorate brain functions [4,117].

## 4. Experimental Studies and Gut-Based Therapy: From Rodents to Humans

Beyond quantitative changes, qualitative disturbances of the normal microbiota occur with chronic alcohol consumption. Specific compositional changes in AH-associated microbiota are a crucial aspect of this disease spectrum and can also complicate brain function. Recent preclinical and clinical studies focused on the improvement in intestinal barrier integrity to provide amelioration in alcohol-induced liver damage. Intervention with conventional methods such as probiotics, prebiotic, antibiotics, and new techniques such as fecal microbial transplantation are already revealing promising outcomes in preclinical and clinical model of ALD [6,118]. A summary of such compositional and concurrent functional studies of ALD and the CNS in animals and human is listed in Table 1 and Table 2.

Apart from these studies, there are growing numbers of studies that focus on the modulation of the gut microbiome through probiotic treatment and fecal transplant in order to treat anxiety, depression, and other forms of mental illness in humans [136,137]. This signifies that modulatory treatment of the gut microbiome or abstinence from alcohol may have beneficial effects on AUDs by targeting the gut–liver–brain axis. Withdrawal of alcohol improved systemic inflammation by decreasing LPS and other proinflammatory cytokines, thus enhancing mood and cognition and while decreasing alcohol dependence in AUD patients [65]. Beneficial metabolite supplementation, including SCFAs, is also recognized as potential substrates could transform the gut environment, alleviating inflammatory progression.

In an in vitro study, Lesley et al. hypothesized that propionate protects the BBB from oxidative stress induced by LPS via increased expression of free fatty acid receptor-3 in brain epithelium cells [138]. Another in vitro study demonstrated that higher portal propionate levels lowered liver triglyceride content via decreased de novo lipogenesis [139]. Propionate is a one of the major SCFAs that has functional mechanisms in the activation of gluconeogenesis, thereby regulating food intake, enhancing insulin sensitivity, and maintaining metabolic homeostasis in the host’s gut. Increase in bacterial-derived propionate has also shown the ability to regulate metabolic homeostasis and maintain inflammatory markers at low levels without being cytotoxic to liver cells.

In line with this, butyrate supplementation perturbed and mitigated LPS-induced severe inflammation and reshaping of the gut microenvironment in vivo. Moreover, butyrate supplementation increased the abundance of other commensal bacteria, *Faecalibacterium* and *Lactobacillus* [140], which add on to the effects of butyrate supplementation as *Faecalibacterium* are documented butyrate producers that are decreased in ALD patients [27]. Additionally, long term use of dietary supplements containing the *Lactobacillus* and *Bifidobacterium* species was discovered to enhance cognitive and memory functions by altering brain metabolites [141], such as GABA, which regulates glutamine/glutamate signaling in brain. Beneficial bacterial supplementation also increased myo-inositol, which is linked with astrocyte activity and is reduced in brain injury or aging [142]. Identifying new biomarkers influencing brain behavior that predict the susceptibility for addiction to alcohol could be beneficial for clinical therapy in patients with AUDs. In such instances, an altered microbial composition associated with expansions in the dopamine 1 receptor and reductions in the dopamine 2 receptor in the dorsal striatum could be assessed as a proxy for susceptibility to alcohol addiction [143].

Advancements and innovation in functional brain mapping have expanded our insight and could divulge how alcohol modifies the brain on a mechanistic level. Mechanisms by which alcohol triggers neuro-inflammation are starting to be disentangle as alcohol and its metabolic products have been shown to modify neurotransmitter signals in the brain, including GABA, glutamate, acetylcholine, dopamine, and serotonin [144]. A magnetic resonance spectroscopy study in AC patients with HE found a significantly higher mean glutamate/glutamine to creatinine ratio together with a reduced choline to creatinine ratio when compared with healthy controls [145].

Cervical vagus nerve stimulation could be a novel approach to regulate gut–brain communication at nerve interfaces that affect cognition. An animal study revealed a reduction of LPS-induced systemic and brain inflammation as well as showed significantly enhanced cognitive responses via vagus nerve stimulation [146]. In addition, the beneficial effects of *Lactobacillus rhamnosus* JB1 on neuropsychiatric behavior are prevented after vagotomy [147]. Probiotic *L. reuteri*, in an in vivo study of autism, provided evidence of improvement in social behavior and wound healing that is dependent on vagus-nerve-regulated pathways [148].

Furthermore, clinical trials by Ahluwalia et al. found that more brain edema, hyperammonemia, and significant cortical damage with lower brain reserve was observed in AC patients in comparison with that in cirrhotic patients who were alcohol abstinent for more than 6 months [149]. These complications form a major burden from a medical and psycho-social perspective, and strategies to improve their prognosis have largely depended on gut microbial modulation [150].

**Table 2 jcm-10-00541-t002:** Summary of effects of gut microbe and its metabolites implicated in brain and liver in alcohol-related clinical study.

Model/Disease	Intervention	Effect on Brain and Gut	Effect on Gut and Liver	Ref.
HE	Lactulose withdrawal	↑ Glutamine + Glutamate ↓ Cognitive performance	↓ *Faecalibacterium* ↓ Veillonellaceae	[151]
Cirrhosis and minimal HE	Rifaximin	↑ Cognitive performance ↓ Permeability	↑ Beneficial metabolites ↑ Eubacteriaceae	[152]
Alcoholic cirrhosis	-	Abnormal T1 Weighted hyperintensity in the globi pallidi ↑ Hepatocerebral degeneration	-	[153]
Cirrhosis and minimal HE	Nutritional therapy	-	Significant improvement in MHE, ammonia, MELD, CTP SIP6 score	[154]
Chronic liver diseases including alcoholic cirrhosis Cirrhosis with HE	Probiotic VSL#3	-	Improved MDA and 4-HNE Improved proinflammatory cytokines in AC patients Improved AST, ALT, GGT in AC patients Improved S-NO plasma level in AC patients Improved MELD, CTP score	[155,156]
Cirrhosis and minimal HE	*L. acidophilus*	↓ Glutamine + glutamate/creatinine ratio ↑ Myo-inositol/creatinine ratio ↑ Choline + myo-inositol/creatinine ratio Improved neurometabolites and psychometric analysis	Improved ammonia in blood	[157]

Abbreviations: ↑ indicates an increase in the condition or level; ↓ indicates a decrease in condition or level; HE, hepatic encephalopathy; ALT, alanine transaminase; AST, aspartate aminotransferase; GGT, γ-glutamyl transferase; AC, alcoholic cirrhosis; MDA, malondialdehyde; 4-HNE, 4-hydroxynonenal; S-NO, S-nitrosothiols; MHE, minimal hepatic encephalopathy; MELD, model for end-stage liver disease; CTP, the Child–Turcotte–Pugh; SIP6, Sickness Impact Profile.

## 5. Future Perspective

Given the role of microbiota and intestinal permeability in AUD patients, it would be wise to study the gut microbiota as a potential target for reducing the mediating effects of alcohol. Additionally, while alcohol is most certainly the driver of events in the gut and liver AUDs [34], it also clearly has an unsettling relation with the brain [97]. Latest advancements in brain mapping, neuro-imaging, high-throughput DNA sequencing, and improved computational techniques have broadened our approach in understanding how alcohol influences the brain, gut, and liver [4,158]. However, difficulties arise as the impact of the microbial environment in the host is continual and therefore difficult to parse under some conditions.

In the last two decades, various studies have determined the impact of microbiota and their toxic metabolites in various neurological disorders and neuroinflammation as well as the interplay with the ENS [159]. More thorough studies are required to fully comprehend the gut–liver–brain links in ALD. By obtaining adequate information, a better approach may be devised from a mechanistic perspective for a prognosis that can be tested to clarify and validate the severity of disease. With alcohol being highly addictive, withdrawal behavior could be abated through manipulation of the gut microbiome and abstinence [160]. Moreover, the possible role of the vagal pathway in alcohol dependence has not yet been investigated comprehensively. Research in actively drinking and sober alcoholics should be performed to the determine the depth of vagal signaling, its influence on brain function, its influence on behavior in alcoholism, and its plausible interaction with systemic inflammation [29].

Considering the complex neuronal and humoral factors in the gut–liver–brain axis, a systemic approach is needed to unravel the downstream pathways that instigate pathological changes in the gut, brain, and liver. Furthermore, studies on patients with cirrhosis and current alcohol use are sparse and limited due to ethical concerns. Patients with alcoholic cirrhosis are often included in cirrhosis clinical trials, but the amount or duration of past or current alcohol use are rarely described.

## 6. Conclusions

Gut microbiome research is rapidly expanding, due to recent advances in high-throughput omics technologies, and providing us with a better understanding of the composition and functionality of such a complex ecosystem. Compelling evidence suggests alcohol abuse is directly linked with gut microbiota dysbiosis and the production of toxic metabolites.

Additional investigation of bacterial metabolites and their potential interactions in gut–liver–brain axis immune signaling, pathways, and neuronal function will enable a fuller understanding of the physiological responses to alcohol and microbiome. Despite our current scant knowledge of specific mechanisms, humoral, neuronal, and microbial modulation are proving promising strategies to tackle alcoholic-disease-associated neurological disorders. A deeper understanding of gut microbial ecology, metabolism, and signaling networks within the host may lead to a new generation of microbiome-targeted strategies, both for disease treatment and prevention.

## Figures and Tables

**Figure 1 jcm-10-00541-f001:**
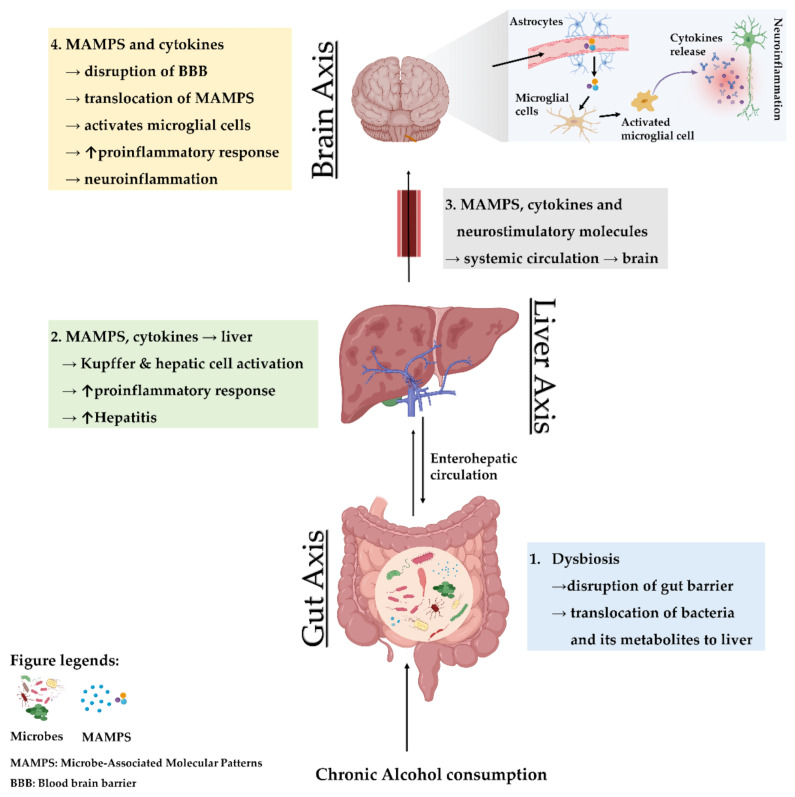
Gut–liver–brain axis in alcohol use disorder. Abbreviations: ↑ indicates an increase in the condition or level; MAMPS, microbe-associated molecular patterns; BBB, blood–brain barrier.

**Table 1 jcm-10-00541-t001:** Summary of effects of gut microbe and its metabolites implicating in brain and liver in alcohol-related animal study.

Model/Disease	Intervention	Effects on Brain and Gut	Effect on Gut and Liver	Ref.
20% ethanol ad libitum in Wistar, Long-Evans and Sprague–Dawley rats	Sodium butyrate MS-275	↓ Alcohol intake	-	[119]
Traumatic brain injury on C57Bl/6 mice	Sodium butyrate	↑ Brain TJ proteins expression ↓ Brain permeability	-	[85]
Male ICR mice received 10% DMSO for 2 weeks followed by single binge of 50% *v*/*v* of ethanol	Kaempferol	-	↑ TJ proteins expression ↓ AST and ALT ↑ Butyrate receptor and transporter protein expression ↓ Hepatic inflammation	[120]
DNBS solution in 30% ethanol injected intrarectally in male C57BL/6 mice	*F. prausnitzii*	↓ Colonic serotonin level	-	[121]
Chronic-binge ethanol feeding in C57Bl/6 female mice	Synbiotic (*F. prausnitzii*, potato starch)	-	↑ TJ proteins expression ↓ Permeability ↓ Hepatic inflammation	[122]
α-synuclein overexpressing germ-free BDF1 mice	Acetate/propionate/butyrate	↑ Microglia activation	-	[123]
38% for 2 weeks, 46% for 3 weeks 56% Ethanol for 3 weeks per day intragastrically to Kunming mice.	Dietary okra seed oil	-	↑ Propionate/butyrate ↓ Intestinal dysbiosis ↓ Hepatic inflammation ↓ Hepatic lipid accumulation	[124]
male BALB/c	*L. rhamnosus*	↓ Corticosterone level ↓ Anxiety- and depression-related behavior	-	[125]
50–60% ethanol (4 g/kg) twice daily dose to male Sprague–Dawley rats	*L. rhamnosus*	-	↓ Oxidative stress ↓ Colonic MPO level ↓ Hepatic inflammation ↓ Permeability	[126]
APP/PS1-Tg C57BL/6 mice	FMT transplantation from AD patients	↑ Intestinal NLRP3 inflammasome response ↑ Cognitive dysfunction ↑ Microglia activation	-	[127]
Female C57Bl/6 wild-type, P2rx7-KO, ssUOX-Tg, intUOX-Tg fed Lieber-DeCarli ethanol diet	Reduced inflammasome activation	-	↓ Uric acid ↓ ATP signaling ↓ Steatosis and hepatic triglyceride level	[128]
Rat cortical astrocytes	Ethanol-induced TLR4/IL-1RI signaling	↑ TLR4 and/or IL-1RI activation ↑ Astrocyte cell death ↑ NF-κB and AP-1	-	[129]
C57BL/6 wild-type mice and TLR4^−/−^ mice	Ethanol (4 g/kg) for 3 days in TLR4^−/−^ mice vs. wild type	↓ Microglia activation	-	[130]
C57BL/6 wild-type mice and TLR4^−/−^ mice fed Lieber–DeCarli ethanol diet	Lieber–DeCarli ethanol diet in TLR4^−/−^ mice vs. wild type	-	↓ ALT level ↓ Hepatic inflammation ↓ Oxidative stress	[131]
C57BL/6 rtTA, and rtTA-Egfr*Tg mice fed Lieber–DeCarli ethanol diet	*L. plantarum*	↓ Systemic inflammation ↓ Neuroinflammation ↓ Gut dysbiosis	-	[132]
C57BL/6J mice fed with LA101A ethanol diet for 6 weeks	*L. plantarum*	-	↓ ALT and AST level ↓ Hepatic inflammation and endotoxin ↓ Oxidative stress ↑ TJ proteins expression	[133]
Wild-type C57BL/6 female mice fed Lieber–DeCarli ethanol diet	Antibiotic cocktail: Ampicillin, Neomycin, Metronidazole, and Vancomycin	↓ Neuro and systemic inflammation ↓ Microglia activation ↓ LPS and bacterial load	-	[134]
Wild-type C57BL/6 female mice fed Lieber–DeCarli ethanol diet	Antibiotic cocktail: Ampicillin, Neomycin, Metronidazole, and Vancomycin	-	↓ LPS and bacterial load ↓ Hepatic inflammation ↓ MPO ↑ Hepatic steatosis	[135]

Abbreviations: ↑ indicates an increase in the condition or level; ↓ indicates a decrease in condition or level; ALT, alanine transaminase; AST, aspartate aminotransferase; MS-275, histone deacetylase inhibitor; TJ, tight junction; ICR, Institute of Cancer Research; DMSO, dimethyl sulfoxide; BDF1, B6D2F1 [C57BL/6 × DBA/2) F1] mice; MPO, myeloperoxidase; APP/PS1, human amyloid precursor protein/presenilin 1; Tg, transgenic; FMT, fecal microbial transplantation; AD, Alzheimer’s disease; NLRP3, nucleotide-binding oligomerization domain-like receptors (NLR) family pyrin domain containing 3; P2rx7, ATP receptor 2 × 7 KO, knock out; intUOX, unmodified intracellular uricase; ssUOX, secreted uricase; ATP, adenosine triphosphate; TLR4, toll-like receptor 4; IL-1RI, interleukin 1 receptor, type I; NF-κB, nuclear factor kappa-light-chain-enhancer of activated B cells; AP-1, activator protein 1; rtTA, reverse tetracycline-controlled transactivator; Egfr, epidermal growth factor receptor; LPS, lipopolysaccharide.

## Data Availability

Not applicable.

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
