# Peer review of "Gut Microbiota at the Intersection of Alcohol, Brain, and the Liver"

_jcm, 2021, doi:10.3390/jcm10030541_

Round 1

Reviewer 1 Report

Liver diseases (hepatopathies) vary in severity, ranging from mild and reversible steatosis to non-alcoholic fatty liver disease (NASH), hepatitis, fibrosis, cirrhosis and, in extreme cases, hepatocellular carcinoma (liver cancer). It seems that these risk factors are not sufficient to lead to liver disease by itself. The disturbance of the intestinal microbiota is also important. Moreover, gut microbiome can be associated with obesity and inflammatory gastrointestinal disorders. Therefore, its modification with prebiotics, probiotics and fecal flora transplantation is a promising field of research in the prevention and treatment of liver diseases, as evidenced by the fact that dysbiosis and alteration of the permeability of the intestinal barrier have been found in all patients with liver disease, regardless of its cause. The more severe the liver damage, the greater the dysbiosis. The gut microflora and its metabolites have been shown to play a role in altering the function of the enteric nervous system (ENS). There is a rapidly growing number of data that links the gut microbiome to the development and functioning of the central nervous system, which is a currently proposed paradigm shift in neuroscience. The strongest evidence for the role of microbes in the gut-brain axis comes from animal studies. According to the recent studies, there is a link between the composition of gut microbiota and mental disorders in animals (response to depression and chronic stress). This phenomenon is known as “gut-brain axis”. This subject requires further examination, especially taking into consideration potential therapeutic options.

The work submitted to me for review is generally very current, interesting and worth considering. The authors present a very extensive review of the current literature, appropriately selected, However in this version, the content of the work is chaotic and hardly understandable. The manuscript requires re-checking. The text is repeated in many places. This makes the content unclear and boring. Needs redrafting.

Author Response

jcm-1079010

“Gut Microbiota at the Intersection of Alcohol, Brain and the Liver”

Point-to-point responses to comments by the Reviewer 1

First of all, we would like to thank the Reviewers 1 for his/her comments, which helped us to improve this manuscript.

Comment: Liver diseases (hepatopathies) vary in severity, ranging from mild and reversible steatosis to non-alcoholic fatty liver disease (NASH), hepatitis, fibrosis, cirrhosis and, in extreme cases, hepatocellular carcinoma (liver cancer). It seems that these risk factors are not sufficient to lead to liver disease by itself. The disturbance of the intestinal microbiota is also important. Moreover, gut microbiome can be associated with obesity and inflammatory gastrointestinal disorders. Therefore, its modification with prebiotics, probiotics and fecal flora transplantation is a promising field of research in the prevention and treatment of liver diseases, as evidenced by the fact that dysbiosis and alteration of the permeability of the intestinal barrier have been found in all patients with liver disease, regardless of its cause. The more severe the liver damage, the greater the dysbiosis. The gut microflora and its metabolites have been shown to play a role in altering the function of the enteric nervous system (ENS). There is a rapidly growing number of data that links the gut microbiome to the development and functioning of the central nervous system, which is a currently proposed paradigm shift in neuroscience. The strongest evidence for the role of microbes in the gut-brain axis comes from animal studies. According to the recent studies, there is a link between the composition of gut microbiota and mental disorders in animals (response to depression and chronic stress). This phenomenon is known as “gut-brain axis”. This subject requires further examination, especially taking into consideration potential therapeutic options.

The work submitted to me for review is generally very current, interesting and worth considering. The authors present a very extensive review of the current literature, appropriately selected, However in this version, the content of the work is chaotic and hardly understandable. The manuscript requires re-checking. The text is repeated in many places. This makes the content unclear and boring. Needs redrafting.

Response: We agree with the reviewer’s comment and thanks for raising this point. We would extend our gratitude for taking out time to review our manuscript and expressing your concern with regard to the contents of this manuscript. As per the reviewer’s suggestions we have made several changes in the content and writing of this manuscript to ease better understanding for the researchers. Perhaps the revised version of manuscript would be much elaborating yet clear on the gut-brain-liver axis.

Reviewer 2 Report

Gut microbiota serves as the most important contributor to the healthy gut as well in the pathophysiology of various diseases. Extensive alcohol consumption leading to series of hepatocellular injuries and brain damages. But the roles of gut microbiota and its metabolites in the connection of gut-liver axis and brain remained unclear. This review had summarized the pathogenic mechanism linked with the gut-liver-brain axis in the development and progression of brain disorders associated with alcoholic liver disease. The topic is attractive and the structure of the review is clear and reasonable. Overall, this review was well prepared expecting several small questions:

  1. Please proofread the manuscript carefully. For example, in “Secondary bile acids (BAs) and short-chain fatty acids (SCFAs) and are the two major…”, the last “and” should be deleted?
  2. The effect of alcohol on gut microbiota is not fully reviewed. How is the role of disrupted microbial diversity by alcohol in the contribution of liver and brain injury?
  3. Is there any study on microbiota dysregulating circadian rhythm?
  4. Is microbiota correlated with alcohol associated neurodegenerative disorders?

Author Response

jcm-1079010

“Gut Microbiota at the Intersection of Alcohol, Brain and the Liver”

Point-to-point responses to comments by the Reviewer 2

First of all, we would like to thank the Reviewers 2 for his/her comments, which helped us to improve this manuscript.

Comment: Gut microbiota serves as the most important contributor to the healthy gut as well in the pathophysiology of various diseases. Extensive alcohol consumption leading to series of hepatocellular injuries and brain damages. But the roles of gut microbiota and its metabolites in the connection of gut-liver axis and brain remained unclear. This review had summarized the pathogenic mechanism linked with the gut-liver-brain axis in the development and progression of brain disorders associated with alcoholic liver disease. The topic is attractive and the structure of the review is clear and reasonable. Overall, this review was well prepared expecting several small questions:

  1. Please proofread the manuscript carefully. For example, in “Secondary bile acids (BAs) and short-chain fatty acids (SCFAs) and are the two major…”, the last “and” should be deleted?
  2. The effect of alcohol on gut microbiota is not fully reviewed. How is the role of disrupted microbial diversity by alcohol in the contribution of liver and brain injury?
  3. Is there any study on microbiota dysregulating circadian rhythm?
  4. Is microbiota correlated with alcohol associated neurodegenerative disorders?
  5.  

Response: We would extend our gratitude for taking out time to review our manuscript.

Comment 1: Please proofread the manuscript carefully. For example, in “Secondary bile acids (BAs) and short-chain fatty acids (SCFAs) and are the two major…”, the last “and” should be deleted?

Response 1: Thank you for pointing out the error. It was corrected as per reviewer’s comment and further detailed proof reading has been done to rectify such errors and has been updated in the revised manuscript.

Comment 2: The effect of alcohol on gut microbiota is not fully reviewed. How is the role of disrupted microbial diversity by alcohol in the contribution of liver and brain injury?

Response 2: Thank you for raising the concern over the role of disrupted gut microbiota by alcohol in contribution to liver and brain injuries. We have tried to explain how the gut microbial dysbiosis through alcohol consumption made an impact on brain and liver injuries and its consequences to gut leakiness. We have also summarized alcohol based gut microbial disruption and tried to correlate with brain injuries through clinical and preclinical studies in the tabular form for better understanding. However, there are very few studies done on gut-brain-liver axis with alcohol contribution, this makes currently hot area for researchers to work. Moreover, after reviewer’s suggestion, changes have been made and contents were added with reference number 109.

Comment 3: Is there any study on microbiota dysregulating circadian rhythm?

Response 3: Thank you for raising this very important question. There are studies done on circadian rhythms associated with microbial dysbiosis and alcohol. We apologize for our overlooking and we have included this area of research in this manuscript under subheadings:

2.1 Alcohol and gut-liver interaction:

Line 65: Furthermore, disruption of circadian rhythm or mutation in circadian gene distinctly affects intestinal permeability and this alteration in gene expression markedly worsens alcohol-induced gut dysbiosis, hepatic injury and hepatic inflammation as showed in vitro and in vivo studies

Reference number: 61, 62, 63.

2.2. Alcohol and gut-brain interaction:

Line 17: Another aspect influencing gut-brain interaction is central regulatory circadian mechanisms in brain which can alter the circadian clock in gastrointestinal track leading to susceptibility to intestinal physiology. Since alcohol use has unsetting relationship with circadian clock, this further exacerbates alcohol effect on intestinal barrier integrity and has potential role in liver and brain injuries.

Reference number: 61, 73, 74.

Comment 4: Is microbiota correlated with alcohol associated neurodegenerative disorders?

Response 4: Thank you for pointing this question. As the multi-omics studies has been developed few years back, there are studies on related to gut microbiome in neurodegenerative disorders however, there are very less studies related to alcohol-associated neurodegenerative disorders. We have tried to explain in the manuscript under subheading-2.2. Alcohol and gut-brain interaction and focused more on neuroinflammation and cognitive decline due to alcohol and gut microbial dysbiosis which are the key factors for neurodegenerative disorders. Alcohol induced gut dysbiosis related neuroinflammation study was included in tabular data under reference number 141.

Reviewer 3 Report

Dear Dr.

Editor,

Final comments:

 This paper shows microbiota and intestinal permeability in alcohol abuse patients. The authors also described brain damage induced by alcohol. This paper has shown detailed mechanisms of chronic alcohol comsumption.

I think this paper is good for publication in this present form.

Author Response

jcm-1079010

“Gut Microbiota at the Intersection of Alcohol, Brain and the Liver”

Point-to-point responses to comments by the Reviewer 3

First of all, we would like to thank the Reviewers 3 for his/her comments, which helped us to improve this manuscript.

Comment: This paper shows microbiota and intestinal permeability in alcohol abuse patients. The authors also described brain damage induced by alcohol. This paper has shown detailed mechanisms of chronic alcohol consumption.

I think this paper is good for publication in this present form.

Response:

We would extend our gratitude for taking out time to review our manuscript. We certainly hope this manuscript would be able to provide sound knowledge of gut-liver-brain axes intertwined with chronic alcohol use in alcoholic liver disease.

Round 2

Reviewer 1 Report

I appreciate the Authors' efforts, but the form of the proofreading is not easy to decipher. Nevertheless, I accept the substantive form of the manuscript. Editorial proofreading is essential.
I suggest more specificities, less generalities in the future.